# Effects of Red Ginseng on Neural Injuries with Reference to the Molecular Mechanisms

**Pengxiang Zhu *** 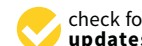 **and Masahiro Sakanaka**

Department of Functional Histology, Ehime University Graduate School of Medicine, Ehime 791-0295, Japan; sakanaka@m.ehime-u.ac.jp

* Correspondence: pxzhu@m.ehime-u.ac.jp; Tel.: +81-89-960-5236

**Abstract:** Red ginseng, as an effective herbal medicine, has been traditionally and empirically used for the treatment of neuronal diseases. Many studies suggest that red ginseng and its ingredients protect the brain and spinal cord from neural injuries such as ischemia, trauma, and neurodegeneration. This review focuses on the molecular mechanisms underlying the neuroprotective effects of red ginseng and its ingredients. Ginsenoside Rb1 and other ginsenosides are regarded as the active ingredients of red ginseng; the anti-apoptotic, anti-inflammatory, and anti-oxidative actions of ginsenosides, together with a series of bioactive molecules relevant to the above actions, appear to account for the neuroprotective effects in vivo and/or in vitro. Moreover, in this review, the possibility is raised that more effective or stable neuroprotective derivatives based on the chemical structures of ginsenosides could be developed. Although further studies, including clinical trials, are necessary to confirm the pharmacological properties of red ginseng and its ingredients, red ginseng and its ingredients could be promising candidate drugs for the treatment of neural injuries.

**Keywords:** red ginseng; ginsenoside Rb1; ischemia; neurotrauma; neurodegeneration; Bcl-2; NF-κB; Nrf2

## 1. Introduction

As a variety of traditional medicine, ginseng root (Panax ginseng CA Meyer) has been utilized for centuries and is one of the most trusted herbs in Asia. There are two types of ginseng products: red ginseng (RG), which is steamed and dried ginseng root as described in the Japanese Pharmacopoeia, and white ginseng (WG), which is dried but not steamed. Following vapor steaming, RG can not only be preserved for a long time but also endowed with new ingredients not present in fresh ginseng. Many studies have demonstrated that RG and its ingredients show broad pharmacological effects in human and animal models. These effects include anti-aging, anti-oxidant, anti-inflammation, anti-diabetes, anti-depression, among others [1–3].

Injury to the nervous tissue is commonly caused by ischemia, trauma, and neuro-degeneration [4]. Historically, RG was thought to be beneficial to the nervous system, and modern researchers have shown increasing evidence for the favorable effects of RG and its ingredients on the nervous system under pathological conditions [5]. This review aims to summarize the effects of RG and its ingredients on neural injuries and to discuss the possible molecule mechanisms underlying the neuroprotective effects of RG, thereby raising new possibilities for the treatment of neural injuries.

## 2. The Main Active Ingredients of Red Ginseng (RG)

The main active components of ginseng are ginsenosides, a special group of triterpenoid saponins [6]; there are more than 100 kinds of ginsenosides which have been isolated from the *panax* species. Protopanaxadiol (PPD) and protopanaxatriol (PPT) are the main structures of ginsenosides (Figure 1) [7]. Some studies have characterized the chemical compositions of RG and WG, indicating

the differences between RG and WG at the molecular level [8–10]. Ginsenosides in RG are known to increase in content and new ginsenosides can be generated within RG with steaming (Table 1) [10,11]. We investigated an RG extract with high-performance liquid chromatography (HPLC) to ascertain the presence of the main ginsenosides as the active molecules in RG (Figure 2) [12].

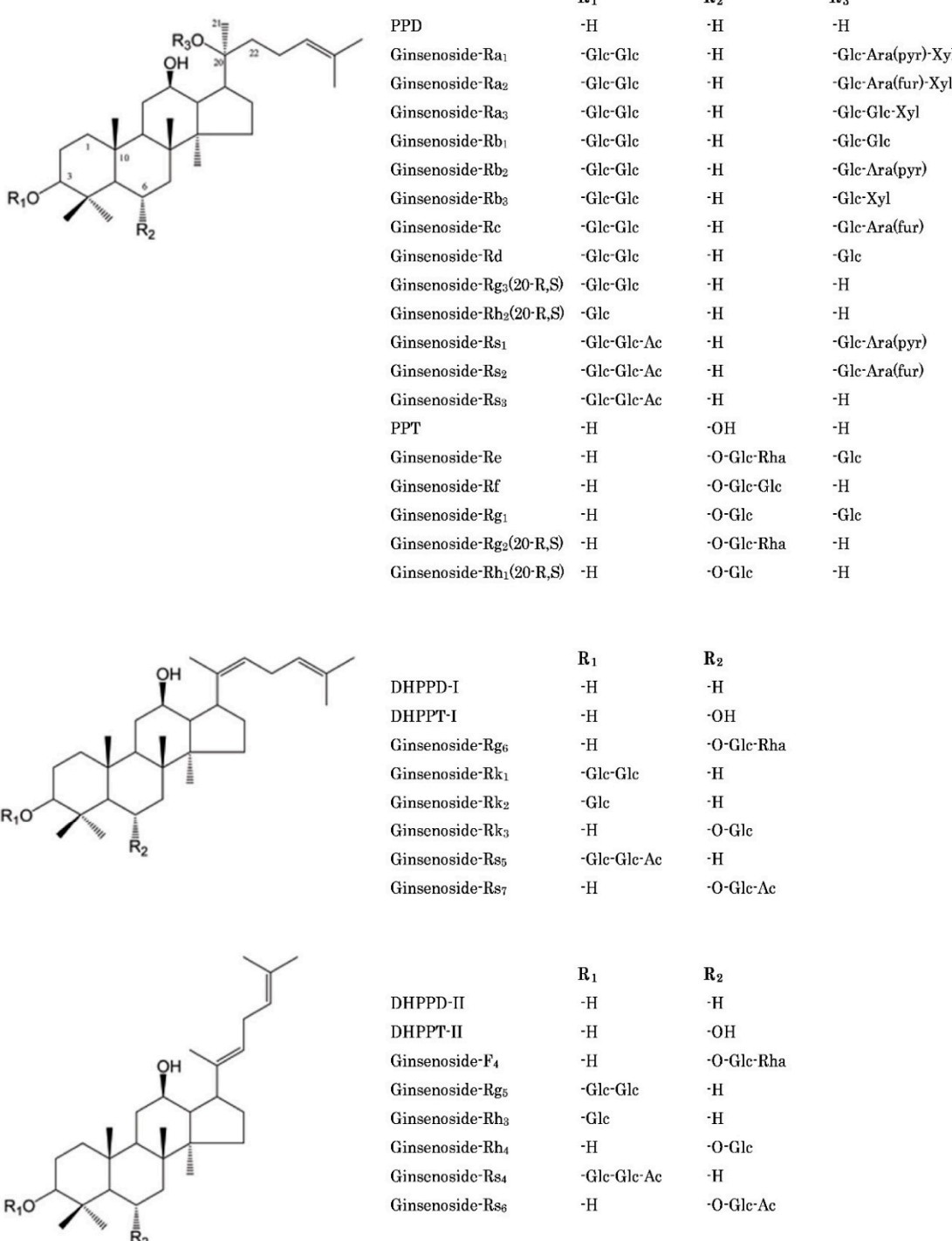

|  | $R_1$ | $R_2$ | $R_3$ |
|---|---|---|---|
| PPD | -H | -H | -H |
| Ginsenoside-Ra$_1$ | -Glc-Glc | -H | -Glc-Ara(pyr)-Xy |
| Ginsenoside-Ra$_2$ | -Glc-Glc | -H | -Glc-Ara(fur)-Xyl |
| Ginsenoside-Ra$_3$ | -Glc-Glc | -H | -Glc-Glc-Xyl |
| Ginsenoside-Rb$_1$ | -Glc-Glc | -H | -Glc-Glc |
| Ginsenoside-Rb$_2$ | -Glc-Glc | -H | -Glc-Ara(pyr) |
| Ginsenoside-Rb$_3$ | -Glc-Glc | -H | -Glc-Xyl |
| Ginsenoside-Rc | -Glc-Glc | -H | -Glc-Ara(fur) |
| Ginsenoside-Rd | -Glc-Glc | -H | -Glc |
| Ginsenoside-Rg$_3$(20-R,S) | -Glc-Glc | -H | -H |
| Ginsenoside-Rh$_2$(20-R,S) | -Glc | -H | -H |
| Ginsenoside-Rs$_1$ | -Glc-Glc-Ac | -H | -Glc-Ara(pyr) |
| Ginsenoside-Rs$_2$ | -Glc-Glc-Ac | -H | -Glc-Ara(fur) |
| Ginsenoside-Rs$_3$ | -Glc-Glc-Ac | -H | -H |
| PPT | -H | -OH | -H |
| Ginsenoside-Re | -H | -O-Glc-Rha | -Glc |
| Ginsenoside-Rf | -H | -O-Glc-Glc | -H |
| Ginsenoside-Rg$_1$ | -H | -O-Glc | -Glc |
| Ginsenoside-Rg$_2$(20-R,S) | -H | -O-Glc-Rha | -H |
| Ginsenoside-Rh$_1$(20-R,S) | -H | -O-Glc | -H |

|  | $R_1$ | $R_2$ |
|---|---|---|
| DHPPD-I | -H | -H |
| DHPPT-I | -H | -OH |
| Ginsenoside-Rg$_6$ | -H | -O-Glc-Rha |
| Ginsenoside-Rk$_1$ | -Glc-Glc | -H |
| Ginsenoside-Rk$_2$ | -Glc | -H |
| Ginsenoside-Rk$_3$ | -H | -O-Glc |
| Ginsenoside-Rs$_5$ | -Glc-Glc-Ac | -H |
| Ginsenoside-Rs$_7$ | -H | -O-Glc-Ac |

|  | $R_1$ | $R_2$ |
|---|---|---|
| DHPPD-II | -H | -H |
| DHPPT-II | -H | -OH |
| Ginsenoside-F$_4$ | -H | -O-Glc-Rha |
| Ginsenoside-Rg$_5$ | -Glc-Glc | -H |
| Ginsenoside-Rh$_3$ | -Glc | -H |
| Ginsenoside-Rh$_4$ | -H | -O-Glc |
| Ginsenoside-Rs$_4$ | -Glc-Glc-Ac | -H |
| Ginsenoside-Rs$_6$ | -H | -O-Glc-Ac |

**Figure 1.** Chemical structures of ginsenosides in Panax ginseng, reproduced with permission from Baek, S.-H, et al. [7]. PPD: protopanaxadiol; PPT: protopanaxatriol; DH: dehydro; Glc: glucose; Ara: abrabinose; Xyl: xylose; Rha: rhamnose; Ac: acetyl; pyr: pyranosyl; fur: furanosyl.

**Table 1.** Variation of ginsenoside content (mg/g) in ginseng root during the steaming process, reproduced with permission from Ki-Yeul, N. [10]. ↑: increased. *: Newly produced.

|  | Steaming Time (Hours) | | | |
|---|---|---|---|---|
|  | 0 | 1 | 2 | 3 |
| Total | 35.439 | 42.113↑ | 57.974↑ | 50.332↑ |
| Diol | 21.768 | 28.488↑ | 43.063↑ | 36.974↑ |
| Triol | 11.569 | 11.505 | 12.533↑ | 11.237 |
| Diol/Triol | 1.867 | 2.476↑ | 3.436↑ | 3.29↑ |
| Ginsenoside-Ro | 2.012 | 2.12↑ | 2.378↑ | 2.121↑ |
| Rb1 | 8.097 | 10.88↑ | 16.016↑ | 14.316↑ |
| Rb2 | 5.531 | 7.23↑ | 11.352↑ | 9.282↑ |
| Rc | 4.989 | 6.495↑ | 9.943↑ | 8.119↑ |
| Rd | 1.251 | 1.629↑ | 2.31↑ | 1.864↑ |
| Re | 4.779 | 4.105 | 5.223↑ | 3.72 |
| Rf | 2.54 | 2.533 | 2.669↑ | 2.413 |
| Rg1 | 3.859 | 4.335↑ | 3.723 | 3.971↑ |
| 20(S)-Ginsenoside Rg1 | 0.481 | 0.468 | 0.7↑ | 0.589↑ |
| 20(R)-Ginsenoside Rg2 | - | - | 0.042 * | 0.125 * |
| 20(S)-Ginsenoside Rg3 | - | 0.093 * | 0.221 * | 0.408 * |
| 20(R)-Ginsenoside Rg3 | - | 0.054 * | 0.14 * | 0.262 * |
| 20(S)-Ginsenoside Rh1 | - | 0.064 * | 0.134 * | 0.277 * |
| 20(R)-Ginsenoside Rh1 | - | - | 0.042 * | 0.142 * |
| Ginsenoside-Rs1 | 1.143 | 1.105 | 1.577↑ | 1.233↑ |
| quinquenoside-R1 | 0.757 | 1.002↑ | 1.504↑ | 1.49↑ |

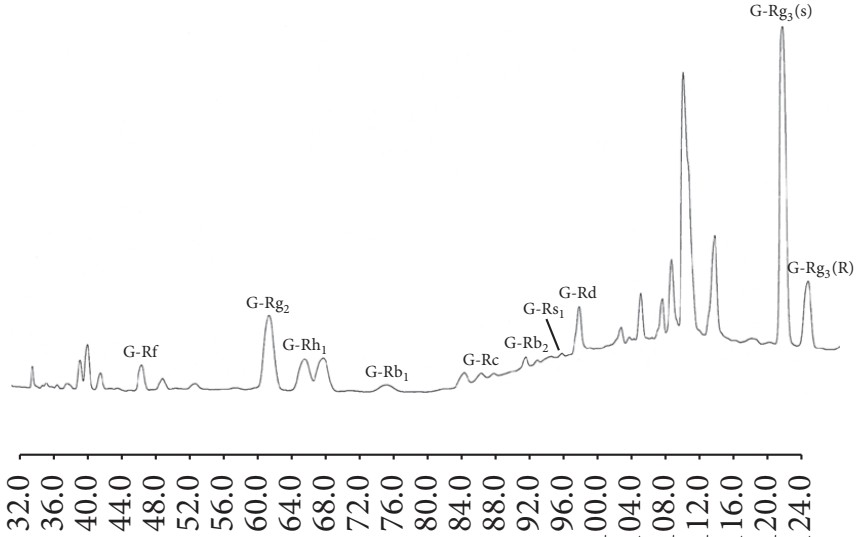

**Figure 2.** HPLC separation of ginsenosides in a red ginseng extract, reproduced with permission from Zhu PX, et al. [12]. Column: Superspher RP-18(e) (4.0 mm i.d. × 250 mm, Merck); mobile phase: (A) $CH_3CN-H_2O-0.1\%\ H_3PO_4$ (21:72:8, v/v), (B) $CH_3CN$; flow rate: total flow 0.8 mL/min (flow program: (A) 0 →19 min: 100%; 19 → 20 min: 100 → 90%; 20 → 73 min: 90%; 73 →103 min: 90→70%; 103→120 min: 70%); Temp.: Temp. program (0→30 min: 35 °C; 30→60 min: 55 °C; 60→120 min: 35 °C); detection: UV 202 nm.

Additional components, such as flavonoids, polysaccharides, volatile oils, and gintonin, as well as nonsaponin compounds, are also found in RG [13]. Although reports have suggested pharmacological properties of these compounds, more evidence is needed to confirm these effects.

### 3. Effects of RG on Brain Ischemia

When an artery leading to the brain is occluded, insufficient blood supply causes cerebral hypoxia and subsequent damage to nerve cells in situ. This pathological condition is known as brain ischemia. The injured area with irreversible damage is termed the ischemic core, and the surrounding tissue is called the ischemic penumbra. Because of the collateral circulation, the penumbra is salvageable during a short period after ischemia and can become a target for the treatment of acute ischemic stroke [14]. Effective treatment for ischemic stroke should include resuming the blood supply as soon as possible and keeping the penumbra viable until and after the blood supply is regained. Intravenous administration of recombinant tissue plasminogen activator(r-tPA) is the only treatment for acute ischemic stroke that has been approved by the U.S. FDA [15]. Numerous studies have been conducted to search for effective neuroprotective compound(s) or drug(s) in the last 30 years. RG was reported to exhibit neuroprotective effect by Wen et al. [16]. They showed that RG powder prevent learning disabilities and neuronal loss in gerbils with 5-minute forebrain ischemia. In this animal model, both orally administered RG powder and intraperitoneally injected crude ginseng saponin significantly protected hippocample CA1 neurons against delayed neuronal death. The neuroprotective molecule in RG powder was identified as ginsenoside Rb1(gRb1) on the basis of in vitro and in vivo studies [16,17]. Zhang et al. proved the neuroprotective action of gRb1 using spontaneous hypotensive rats with permanent occlusion of a unilateral middle cerebral artery distal to the striate branches [18]. In this study, gRb1 was administered directly into the left lateral ventricle with an osmotic minipump to avoid the obstacle of the blood-brain barrier; as a result, gRb1 significantly reduced the infarct area within the temporoparietal cortex. Another active molecule in RG, ginsenoside Rh2, was reported to have a protective effect on ischemia-reperfusion cerebral injury in rats [19]. Ginsenoside Rh2 is a breakdown product of gRb1, and is increased in content when RG is fermented [20]. Intravenous infused gRb1 was reported to decrease infarct volume in the same ischemia model as used in the study of Zhang et al. [18], as well as ameliorated the space navigation ability of the rats with cerebral infarction [21]. This study demonstrated the anti-apoptotic effect of gRb1, together with a molecular mechanism involving Bcl-xL and its promotor STAT5. The Bcl-xL protein is a member of the Bcl-2 family and has anti-apoptotic activity by suppressing the release of cytochrome c [22]. Upregulation of Bcl-xL expression in neurons around ischemic penumbra has been reported [23], and overexpression of Bcl-xL is known to diminish the infarct volume in mice with permanent occlusion of the unilateral middle cerebral artery (MCAO) [24]. These studies proved that Bcl-xL is involved in the apoptotic processes caused by ischemic stroke, and that upregulation of Bcl-xL expression could prevent the apoptotic processes, thus rescuing a population of neurons from ischemic damage. A stable chemical derivative of gRb1, dihydroginsenoside Rb1 (dgRb1), was also reported to alleviate ischemic brain damage through upregulation of Bcl-xL and VEGF when infused intravenously [25]. The effective dose of dgRb1 was ten times lower than that of gRb1, possibly because dgRb1 is chemically more stable than gRb1 and thus less easily bio-degradable (Figure 3). Another Bcl-2 family member, Bcl-2, was also reported to be upregulated by gRb1 and ginsenoside Rh2 in ischemic animal models [19,26,27]. Cells initiating apoptosis show an increase in cytosolic cytochrome c, which converts pro-capase-3 into active caspase-3 to induce apoptosis. The Bcl-2 family could prevent this upregulation of cytochrome c, thereby interrupting apoptosis [28]. Several studies have reported that cytochrome c and caspase-3 may be inhibited by gRb1 administration in vivo and in vitro [29,30]. In summary, RG and its active components prevent neuronal apoptosis through the interruption of the caspase cascade (Figure 4). Furthermore, gRb1 appears to upregulate the expression of certain neurotrophic factors, such as BDNF and GDNF, during transient MCAO in rats [27,30]. These neurotrophic factors are necessary for the growth, maintenance, and survival of cells in the brain, and thus they are likely to protect neurons from ischemic stroke.

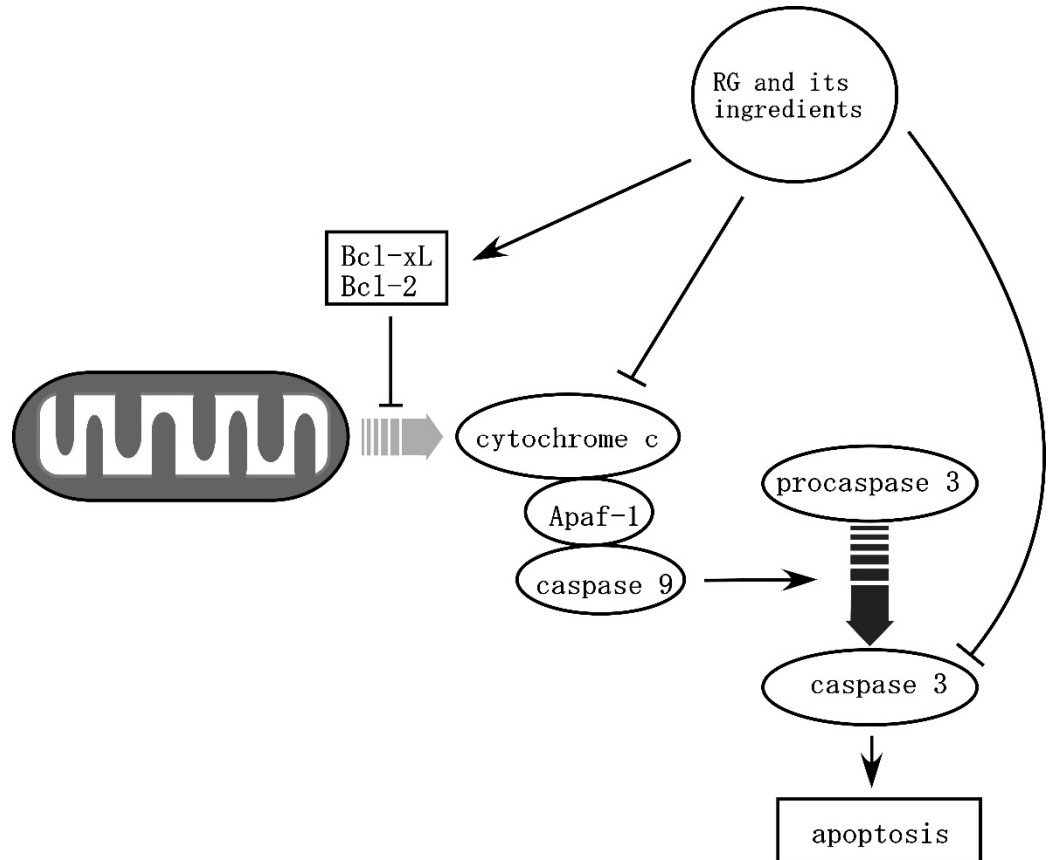

**Ginsenoside Rb1 (gRb1)**          **Dihydroginsenoside Rb1 (dgRb1)**

**Figure 3.** Molecular structures of gRb1 and dgRb1. The arrow indicates the difference between gRb1 and dgRb1, reproduced with permission from Sakanaka, M. et al. [25].

**Figure 4.** Diagram of the effect of RG and its ingredients on the caspase cascade.

In addition to its anti-apoptotic effect, RG and its ingredients have been reported to inhibit inflammatory responses after brain ischemia. Ischemic insult to the brain immediately induces the response of microglia and the secretion of pro-inflammatory cytokines, which makes the ischemic lesion larger as time progresses [31]. Intranasally applied gRb1 reduced infarct volume and alleviated

the neurological severity score in an MCAO rat model by suppressing the activation of microglia in the penumbra; it also reduced the expression of pro-inflammatory cytokines, such as TNF-$\alpha$ and IL-6 [32]. Nuclear factor-$\kappa$B (NF-$\kappa$B) is a protein complex which plays a key role in immune responses [33]. In focal cerebral ischemia, NF-$\kappa$B appears to be activated to promote cell death [34]. Administration of gRb1 has been shown to inhibit the activation of NF-$\kappa$B, to suppress its binding to DNA and to decrease the TNF-$\alpha$ and IL-6 levels, thereby preventing neuron death [32].

A goal of treatment for brain ischemia is the restoration of the cerebral blood supply, which is called reperfusion. Reperfusion after ischemia can bring oxygen and glucose to the brain; however, rapid reperfusion could induce unexpected injuries to the brain. One mechanism underlying ischemia/reperfusion injuries is oxidative stress [35]. An excess of reactive oxygen species (ROS) such as peroxides and free radicals could directly damage the DNA, RNA, protein, and lipids in neurons, inducing neural dysfunction. The nuclear factor erythroid 2-related factor 2 (Nrf2) is a key molecule in the control of cellular anti-oxidative responses [36]. When activated by ROS, Nrf2 can translocate to the nucleus and bind to the antioxidant response element (ARE) to regulate gene expressions involved in anti-oxidation. Pre-activation of Nrf2 has been shown to reduce the liver damage caused by ischemia/reperfusion in mice [37]. Several studies have shown that gRb1 can protect neurons through activation of the Nrf2/ARE pathway in human SK-N-SH dopaminergic cells, SH-SY5Y cells, and neural progenitor cells [38–40]. Shi et al. indicated that gRb1 protected the brain from neural injury through Nrf2 activation, and that the knockdown of Nrf2 counteracted the neuroprotective effect of gRb1 [41]. Liu et al. indicated that gRb1 protected the spinal cord from oxidative stress by activating the Nrf2/ARE signal pathway [42]. Taken together, the neuroprotection achieved by RG and its active ingredients, especially gRb1, is attributed to their anti-apoptotic, anti-inflammatory, and anti-oxidative actions.

## 4. Effects of RG on Neurotrauma

Neurotrauma, including spinal cord injury (SCI) and traumatic brain injury (TBI), is a serious issue that adversely impacts public health around the world. Traumatic injury to the brain or spinal cord causes primary damage to the neurons and axons in situ, and the downstream consequences are known as secondary damage due to ensuing ischemia, inflammatory responses, and excessive release of excitatory neurotransmitters [43,44]. GRb1 and dihydroginsenoside Rb1(dgRb1), as a chemical derivative of gRb1, have been reported to ameliorate compressive spinal cord injury through the upregulation of Bcl-xL and VEGF expression [25,45]. VEGF promotes nerve repair after ischemic stroke or traumatic injury. There are presumably two mechanisms underlying neuroprotection by VEGF: direct neurotrophic action and the angiogenesis-facilitating effect which results in an increase in blood flow within the injured area [46]. Oral administration of an RG extract has also been reported to promote neurorestoration after SCI in rats [12]. The RG extract was shown to prevent neuron loss and to facilitate restoration of white matter, promoting recovery from motor and behavioral abnormalities. Moreover, the RG extract reduced the expression of TNF$\alpha$ and IL-1$\beta$ and decreased the aggregation of Iba-1 positive microglia in the injured spinal cord [12]. Similar studies have reported that RG extract improved functional recovery and neural damage after SCI by inhibiting inflammatory responses [47,48]. In addition, Wang et al. reported that gRb1 treatment decreased neuron loss and promoted functional recovery through inhibition of autophagy following spinal cord injury both in vivo and in vitro [49].

In addition to the neuroprotective effects of RG and its ingredients on experimental animals with SCI, natural products have been shown to exhibit similar favorable effects in the case of TBI. Treatment with ginseng total saponins decreased contusion damage in the cerebrum and improved neurological deficits after experimental TBI [50]. Xia et al. reported that ginseng total saponins attenuated secondary damage after TBI by alleviating oxidative stress in the brain and by downregulating the expression of IL-6 and IL-1β [51]. Another study demonstrated that ginseng ameliorated cognitive deficits after TBI through inhibition of microglia-dependent neuro-inflammation [52]. These studies suggest that RG and its ingredients protect neurons from traumatic damage through suppression of neuroinflammation and neuronal apoptosis.

## 5. Effect of RG on Neurodegeneration

Neurodegeneration is characterized by the progressive loss of neural functions and neurons; it is responsible for motor disorders such as rigidity, ataxias, and/or cognitive impairment. Neurodegenerative diseases, including Parkinson's disease (PD), Alzheimer's disease (AD), Huntington's disease (HD), and amyotrophic lateral sclerosis (ALS), are not completely curable as neurons in the affected lesions fall into apoptosis in a time-dependent manner, regardless of medical treatment. Numerous efforts have been made to find effective therapies, although the etiology of each neurodegenerative disease has not yet been clearly determined. Several studies have shown that RG and its ingredients ameliorate cognitive disorders in AD animal models [53–55]. Aβ and phosphorylated-tau content were also decreased in the brain through use of RG or its ingredients in animal models of AD [55,56]. There are reports that AD patients treated with RG showed significant improvement in cognitive function and an increase in the relative ratio of alpha waves when examined with quantitative electroencephalogram (EEG) [57,58]. Because these clinical studies only included a small number of patients without placebo controls, well-designed and more extensive studies are needed to ascertain the favorable effects of RG on patients of AD [59]. Several RG extracts were shown to have beneficial effects on animal models of PD [60,61] and HD [62]. The putative active molecules of RG available for the treatment of neurodegenerative diseases were identified as the ginsenosides Rb1 and Rg1 [63–65]. In addition to their anti-apoptotic and anti-inflammatory actions, RG and its ingredients were proven to reduce Aβ production in an AD model [53] and to inhibit α-synuclein aggregation in a PD model [66]. Further studies are needed to identify the molecular mechanisms underlying the effects of RG and its ingredients on neurodegenerative diseases.

## 6. Conclusion and Perspectives of RG Research

The studies reviewed above suggest that RG and its ingredients could protect neurons from a variety of injuries, such as ischemia, trauma, and degeneration. Numerous studies have indicated that ginsenoside Rb1 and other ginsenosides are the active molecules responsible for the neuroprotective effects of RG in neural injuries (Table 2). Further studies, including clinical trials, are necessary to confirm the pharmacological properties of RG and its ingredients. Additionally, this review has raised the possibility that more effective or stable neuroprotective derivatives based on the chemical structures of ginsenosides could be developed. In support of this speculation, dihydroginsenoside Rb1 is endowed with a more stable chemical structure compared with gRb1 and was shown to be as effective as gRb1 in terms of neuroprotection at a dose lower than those typically used for gRb1. Overall, RG and its ingredients could be promising candidate drugs for the treatment of neural injuries.

**Table 2.** Neuroprotective effects of RG and its ingredients in neural injury and the molecular mechanisms.

| Types of Neural Injury | Ischemia | Trauma | Neurodegeneration |
| --- | --- | --- | --- |
| Types of study | In vivo and in vitro studies. | In vivo and in vitro studies. | In vivo and in vitro studies, clinical studies. |
| Active components | RG powder, RG extract, gRb1, gRh2, dgRb1 | RG extract, ginseng total saponin, gRb1, dgRb1 | RG powder, gRb1, gRg1 |
| Effect | Prevention of learning disability and neuron loss; reduction of infarct volume. | Prevention of neuron loss and promotion of restoration of white matter after SCI; amelioration of cognitive deficits; decrement of contusion damage after TBI. | Amelioration of cognitive disorders; improvement of cognitive function; increment of relative ratio of alpha waves; reduction of Aβ and phosphorylated-tau; inhibition of α-synuclein aggregation. |
| Proposed mechanisms | Anti-apoptosis: up-regulation of Bcl-xL and Bcl-2; inhibition of cytochrome-c and caspase-3. Neurotrophic action: up-regulation of BDNF and GDNF. Anti-inflammation: inhibition of pro-inflammatory cytokines; inhibition of microglia activation; inhibition of NF-κB activation. Anti-oxidation: activation of Nrf2/ARE pathway. | Anti-apoptosis: up-regulation of Bcl-xL Neurotrophic action: up-regulation of VEGF. Anti-inflammation: inhibition of pro-inflammatory cytokines; inhibition of microglia activation; inhibition of autophagy. Anti-oxidation. | Anti-apoptosis: upregulation of Bcl-xL. Anti-inflammation: inhibition of pro-inflammatory cytokines; inhibition of microglia activation. |
| Reference | [16–21,25–27,29,30,32,38–42] | [12,25,45–52] | [53–58,60–66] |

**Author Contributions:** Conceptualization and writing (original draft preparation), P.Z.; Writing (review and editing), M.S.

**Funding:** This work was funded by JSPS KAKENHI, Grant Numbers: JP16K11407 and JP18K08920.

**Acknowledgments:** The authors appreciate Japan Red ginseng Research Association for providing valuable information of ginseng. The secretarial assistance of Kaori Hiraoka is acknowledged.

**Conflicts of Interest:** The authors declare no conflict of interest.

## Abbreviations

| | |
|---|---|
| RG | red ginseng |
| WG | white ginseng |
| PPD | protopanaxadiol |
| PPT | protopanaxatriol |
| HPLC | high performance liquid chromatography |
| r-tPA | recombinant tissue plasminogen activator |
| gRb1 | ginsenoside Rb1 |
| STAT5 | signal transducer and activator of transcription 5 |
| MCAO | middle cerebral artery occlusion |
| dgRb1 | dihydroginsenoside Rb1 |
| VEGF | vascular endothelial growth factor |
| BDNF | brain-derived neurotrophic factor |
| GDNF | glial cell line-derived neurotrophic factor |
| NF-κB | nuclear factor-κB |
| ROS | reactive oxygen species |
| Nrf2 | nuclear factor erythroid 2-related factor |
| ARE | antioxidant response element |
| SCI | spinal cord injury |
| TBI | traumatic brain injury |
| PD | Parkinson's disease |
| AD | Alzheimer's disease |
| HD | Huntington's disease |
| ALS | amyotrophic lateral sclerosis |
| EEG | electroencephalogram |

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
