# Peer review of "Effects of Red Ginseng on Neural Injuries with Reference to the Molecular Mechanisms"

_2571-8800, doi:10.3390/j2020009_

Round 1
Reviewer 1 Report
This is a useful review that should be published.
Author Response
Response: We have our manuscript edited by MDPI English editing service. The revised manuscript has been resubmitted.
Reviewer 2 Report
Pengxiang Zhu and Masahiro Sakanaka reviewed studies from research about the anti-apoptotic, anti-inflammatory and anti-oxidative actions and molecular mechanisms underlying the neuroprotective effects of red ginseng and its ingredients. Of interest, they suggested that more effective or stable neuroprotective derivatives based on the chemical structures of ginsenosides could be developed. However, the manuscript does not provide a critical assessment about this topic.
In addition, the manuscript is not written very well and it should undergo moderate English editing. I strongly recommend that the manuscript is revised before resubmission.
Author Response
Pengxiang Zhu and Masahiro Sakanaka reviewed studies from research about the anti-apoptotic, anti-inflammatory and anti-oxidative actions and molecular mechanisms underlying the neuroprotective effects of red ginseng and its ingredients. Of interest, they suggested that more effective or stable neuroprotective derivatives based on the chemical structures of ginsenosides could be developed. However, the manuscript does not provide a critical assessment about this topic.
In addition, the manuscript is not written very well and it should undergo moderate English editing. I strongly recommend that the manuscript is revised before resubmission.
Response: We have our manuscript edited by MDPI English editing service. The revised manuscript has been resubmitted.
This manuscript is a resubmission of an earlier submission. The following is a list of the peer review reports and author responses from that submission.
Round 1
Reviewer 1 Report
No new findings or statement of purposes could be found from this work. Many researchers have already studied red ginseng and/or ginsenosides with the explanation of functionalities at molecular level and published a number of high quality review work. There is little value as a review paper. To maintain IJMS journal quality, I disagree publication of this work.
Reviewer 2 Report
In this review, the authors summarized the beneficial effects of red ginseng on neural injuries, with focus on the underlying molecular mechanisms. Three types of neural injuries – brain ischemia, neurotrauma, and neurodegeneration were focused, and primary molecular mechanisms of red ginseng’s effects including anti-apoptosis, anti-inflammation and anti-oxidative action. Overall this review is well organized, and abundant studies were discussed to provide a comprehensive picture of the current research progress on red ginseng’s neural protective potentials.
Main critiques:
1. Resolution of Figure 1 need improvement. In fact, a table providing the list of characterized ginsenosides could work better than a HPLC chromatograph as in Figure 1. The authors can also consider adding white ginseng in the table for comparison.
2. Figure 2. Consider adding more structures of other ginsenosides and RG ingredients discussed in the manuscript.
3. An extra table summarizing the discussed molecular mechanisms can be added. The table can include type of neural injury, type of study (in vivo or in vitro, animal model etc.), type of material (whole RG, RG extract or specific ginsenosides), brief result, and proposed molecular mechanisms.